# Constitutive Modeling of Mechanical Behaviors of Carbon-Based CNTs and GSs, and Their Sensing Applications as Nanomechanical Resonators: A Review

**DOI:** 10.3390/nano13121834

**Published:** 2023-06-09

**Authors:** Toshiaki Natsuki, Jun Natsuki

**Affiliations:** 1College of Textiles and Apparel, Quanzhou Normal University, Quanzhou 362000, China; 2Institute for Fiber Engineering (IFES), Interdisciplinary Cluster for Cutting Edge Research (ICCER), Shinshu University, 3-15-1 Tokida, Ueda 386-8567, Nagano, Japan

**Keywords:** carbon nanotubes, graphene sheets, nanosensor, modeling and simulation, nonlocal elasticity theory

## Abstract

Carbon-based nanomaterials, including carbon nanotubes (CNTs) and graphene sheets (GSs), have garnered considerable research attention owing to their unique mechanical, physical, and chemical properties compared with traditional materials. Nanosensors are sensing devices with sensing elements made of nanomaterials or nanostructures. CNT- and GS-based nanomaterials have been proved to be very sensitive nanosensing elements, being used to detect tiny mass and force. In this study, we review the developments in the analytical modeling of mechanical behavior of CNTs and GSs, and their potential applications as next-generation nanosensing elements. Subsequently, we discuss the contributions of various simulation studies on theoretical models, calculation methods, and mechanical performance analyses. In particular, this review intends to provide a theoretical framework for a comprehensive understanding of the mechanical properties and potential applications of CNTs/GSs nanomaterials as demonstrated by modeling and simulation methods. According to analytical modeling, nonlocal continuum mechanics pose small-scale structural effects in nanomaterials. Thus, we overviewed a few representative studies on the mechanical behavior of nanomaterials to inspire the future development of nanomaterial-based sensors or devices. In summary, nanomaterials, such as CNTs and GSs, can be effectively utilized for ultrahigh-sensitivity measurements at a nanolevel resolution compared to traditional materials.

## 1. Introduction

In various fields of science and technology, nanomaterials, such as carbon nanotubes (CNTs) and graphene sheets (GSs), as well as CNT- and GS-based nanocomposites, have attracted considerable research attention owing to their unique structural, mechanical, and physical properties [1,2,3]. Graphene, also known as monolayer graphite, exhibits single-atom thickness, and a CNT can be regarded as a sheet of graphene rolled into a cylinder. Thus, CNTs and GSs often compete as alternatives to each other in certain applications because of several similarities in their characteristics. Hence, it is advantageous to combine CNTs and GSs and use them together for various applications [4]. Currently, CNTs and GSs have immense potential for designing nanosensing elements, because a nanomechanical resonator is an ultrasensitive sensor with high-frequency oscillation performance that depends on its mass and stiffness. Several studies have revealed that the elastic modulus of ultrasmall carbon-based nanomaterials can attain values as high as 1 TPa, which is stiffer than that of any known substance [5,6]. Owing to the outstanding material characteristics of CNTs and GSs, numerous studies have applied them as sensing elements in various nanosensors and nanodevices.

Nanotechnology has ushered in a new era in materials science and engineering. The research and development of nanomechanical resonator systems include various nanotechnologies, such as fabricating and controlling carbon-based nanomaterials, devising nanodevices, and modeling and computing analysis. The concept of systems integration of materials/devices/analysis for carbon-based nanomechanical resonators is illustrated in Figure 1. With extensive research on nanomaterials, nanoprocessing, and manufacturing technology, the developments of nanotechnology will enable high-sensitivity sensing and detection. To date, several researchers have applied carbon-based nanomaterials as nanosensing elements in biosensors [7,8], gas sensors [9,10], force and strain sensors [11,12,13], and printed CNT electronic sensors [14,15]. The fabrication and evaluation of CNT-derived screen-printed (SP) electrochemical sensors based on a CNT ink were reported [14]. The printed CNT-based flexible electronics allowed for implementation in sensing, providing a promising candidate for such printing processes which can offer stable devices with high performance [15]. Specifically, in CNT-based resonators, progressively high mass-sensing resolution has been achieved using CNTs as the resonating mass detector, realizing atomic-scale detection [16]. The results demonstrated that suspended CNTs can serve as a suitable resonator owing to their low effective mass and acceptable quality factor. Studies on GS-based nanomechanical sensors were conducted using a technique allowing natural frequency decomposition for only nanosheets with nanoparticles affixed to them [17,18]. The fundamental frequencies of triangular, circular, rectangular, and square GSs in equal attached atoms were investigated by modal analysis and compared together [17]. GSs can be utilized as an excellent sensitive balance for detecting nanoparticles.

Over the past two decades, an increasing number of papers and reviews have explored carbon-based nanomaterials, such as CNTs and GSs. In this review, we focus on mathematical modeling and differential governing equations to understand the advanced mechanical behaviors and sensing applications of carbon-based CNTs and GSs. In this regard, the sensing applications primarily included CNT/GS-based nanomass and nanoforce sensors, especially for assessing the working mechanisms on the basis of available theoretical analysis models and simulations. This research reviews and discusses the mechanical properties of carbon-based nanoresonators and their applications in nanomass and nanoforce sensors. First, we introduced the excellent mechanical properties of CNTs/GSs, including the theoretical analysis models that have been employed to simulate and characterize their mechanical properties, such as tensile strain and strength, vibration frequency, and buckling stability. Moreover, we discussed the nanomaterial properties of CNTs/GSs to assess their potential application in CNT/GS-based nanomass and nanoforce sensors. Thereafter, the CNT/GS-based nanoresonators were comprehensively described, including their detection mechanisms, modeling approaches, and major challenges and limitations of their application. Lastly, the conclusions of this review are summarized.

## 2. Mechanical Properties, Vibration, and Buckling of CNTs and GSs

### 2.1. Structure, Modeling, and Properties of CNTs

#### 2.1.1. Measurements of Tensile Properties and Analytical Modeling

As reported in various scientific research fields, the discovery of CNTs was attributed to Sumio Iijima in 1991 [19]. Since then, extensive research activities have been conducted worldwide on CNTs because of their exceptional mechanical, thermal, and electrical properties [20,21,22,23]. The fundamental properties that render CNTs as a promising technology for various future applications include extremely high elastic modulus, strength, and tensile deformation up to a strain of 12% [24]. In addition, CNTs exhibit a perfect geometrical structure and nanoscale structure. Therefore, CNTs are ideal reinforcements for various structures and matrix materials used in composites. Owing to the high mechanical load-carrying capacity of CNTs in nanocomposites, the addition of only 1 wt.% CNTs increased the tensile strength of polystyrene-based composite films by 25% [25,26,27].

The performance of any CNT-based nanocomposites or nanostructures depends on the mechanical characteristics of CNTs. Therefore, the mechanical properties of individual CNTs, such as vibration frequency, buckling stability, strength, and deformation, require further investigation. Generally, the majority of the potential applications of CNTs are based on an understanding of their mechanical performance. Recent studies have involved both experimental measurements and theoretical analyses of the mechanical properties of CNTs [28,29,30,31,32,33,34]. The experimental methods for measuring the mechanical properties are based on the techniques of transmission electron microscopy (TEM) and atomic force microscopy (AFM). Treacy et al. estimated the elastic modulus of individual CNTs by employing an experimental method using TEM [28]. They observed TEM images of CNTs undergoing thermal vibration with its mean-square amplitude that was proportional to the temperature. Furthermore, they measured the amplitude of the intrinsic thermal vibrations, and the elastic modulus of multiwalled CNTs (MWNTs) was evaluated to be 1.8 ± 0.9 TPa. Wong et al. measured the bending force of MWNTs using the tip of AFM and derived a slightly lower value of 1.28 ± 0.59 TPa [29]. Krishnan et al. measured the elastic modulus of single-walled CNTs (SWNTs) in the diameter range of 1.0–1.5 nm. The mean value of 27 nanotubes has been estimated to be ~1.25 TPa [30]. However, the direct measurement of the mechanical properties is remarkably difficult because of the CNTs’ nanoscale dimensions. The experimental results were derived from indirect predictions based on the mechanics of vibration and bending beam theory. In the experimental measurements, the simplified continuum beam theory of CNTs can be utilized to evaluate the elastic modulus, assuming small bending deformations and large aspect ratios. The motion of the CNT beam governed without rotary inertia and shear effects can be expressed as
(1)$$EI\frac{\partial^{4}\;u\left(x,\;t\right)}{\partial x^{4}}+\rho A\frac{\partial^{2}\;u\left(x,\;t\right)}{\partial t^{2}}=p\left(x\right),$$
where *u*(*x*, *t*) denotes the displacement, *x* denotes the axial coordinate, *t* indicates the time, *ρ* denotes the mass density of CNT, *A* indicates the cross-sectional area, *p* represents the distribution of the transverse loading, *E* denotes the modulus of elasticity, and *I* indicates the moment of inertia of the cross-section. 

According to Equation (1), the natural frequency of the CNT beam for the *j*-th vibration mode yields
(2)$$\omega_{j}=\frac{\beta_{j}^{2}}{L^{2}}\sqrt{\frac{EI}{\rho A}},\;\;\;j=1,\;2,\;$$
where *L* denotes the length of the CNT beam, and $\beta_{j}$ indicates the root of an eigenvalue equation dependent on the boundary conditions. For the boundary condition of clamped at one end and free at the other end, this equation can be stated as $\cos\beta_{j}\;\cos h\beta_{j}+1=0$. Thus,$\;\beta_{1}=1.875$, $\;\beta_{2}=4.694$, and $\beta_{3}=7.855$ can be derived as mode values corresponding to the one, two, and three half-waves, respectively.

The elastic modulus of CNTs can be obtained on the basis of the relationship expressed in Equation (1) by indirectly measuring the vibrational frequency and using the classical Euler–Bernoulli beam theory. However, the measurement accuracy of the elastic modulus obtained from the vibrational frequency is strongly affected by the supported boundary conditions. Natsuki et al. [31] developed an accuracy theoretical model, considering nonlocal interactions and boundary condition effects of cantilever support. As depicted in Figure 2, the cantilever boundary condition of the CNT beam was considered to be elastic support instead of fixed support. For the CNT beam with length *L* and diameter *D*, *L*_1_ and *L*_2_ denote the clamped and exposed lengths of the CNT beam, respectively. According to the nonlocal Euler–Bernoulli beam model [32], the governing differential equation for the transverse vibrations of CNTs can be expressed as
(3)$$EI\frac{\partial^{4}w_{1}}{\partial x^{4}}+\rho A\frac{\partial^{2}w_{1}}{\partial t^{2}}-{\left(e_{0}a\right)}^{2}\left(\rho A\frac{\partial^{4}w_{1}}{\partial x^{2}\partial t^{2}}-\frac{\partial^{2}p_{w}}{\partial x^{2}}\right)=p_{w},\;\;\;\;0\leq x\leq L_{1},$$
(4)$$EI\frac{\partial^{4}w_{2}}{\partial x^{4}}+\rho A\frac{\partial^{2}w_{2}}{\partial t^{2}}-{\left(e_{0}a\right)}^{2}\left(\rho A\frac{\partial^{4}w_{2}}{\partial x^{2}\partial t^{2}}\right)=0,\;\;\;\;\;\;L_{1}\leq x\leq L\;$$
where *x* and *t* denote the axial coordinates and time, respectively, $w_{j}\left(x,\;t\right),\;j=1,\;2$ represents the deflection of CNT, *E* denotes the elastic modulus, *I* indicates the moment of inertia of the cross-sectional area *A*, ρ denotes the mass density, $e_{0}a$ represents the nonlocal parameter, considering that $e_{0}$ is suitable for the material, and *a* denotes the characteristic internal length of a C–C bond (0.142 nm), which is the length of the covalent bond between two carbon atoms. The interaction force $p_{w}=-k_{w}w_{1}$ between the CNT and the clamping elastic foundation represents the distributed transverse pressure exerted on the CNT beam under the clamping force. $k_{w}$ denotes the spring constant relative to the stiffness of the elastic foundation materials. 

According to the proposed elastic boundary conditions of the CNT beam, the simultaneous solution of the governing equations of Equations (3) and (4) yields the following [31]:(5)$$M{\left(\omega ,\;E,\;L_{1},\;L_{2}\right)}_{8\times 8}\;\left[\begin{array}{c} A_{1}\\ A_{2}\\ \;\vdots \\ A_{8} \end{array}\right]=0,$$
where *M* denotes an 8×8 matrix with parameters of vibration frequency (ω), elastic modulus (E), and CNT dimension with clamped and exposed lengths $\left(L_{1},\;L_{2}\right)$.

Therefore, the elastic modulus of CNTs can be accurately predicted by measuring the natural frequencies of the CNT beam from the condition of a nontrivial solution of ${\left|M\right|}_{8\times 8}$ stated in Equation (5). The influence of the clamping stiffness at the elastic support end on the accuracy of predicting elastic modulus is displayed in Figure 3. When the length ratio $\left(L_{1}/L_{2}\right)$ of clamping to CNT length was larger than 40%, the elastic modulus of CNTs approached unity as the stiffness of the clamped segment increased. As observed, the elastic modulus was more sensitive to the clamping length of the CNT beam supported by a lower stiffness foundation. Notably, the clamping length and stiffness significantly influenced the resonant frequency of the CNT beam, especially at short clamping lengths.

Moreover, Yu et al. [33] employed a direct measurement method applying a tensile force on CNTs to determine the tensile strength and modulus of single-walled CNTs (SWCNTs) in the ranges of 11–63 GPa and 0.27–0.95 TPa, respectively. These experimental tests both directly and indirectly contributed to confirming the excellent mechanical properties of CNTs. However, they encountered challenges in determining the dependence of CNT structures, such as zigzag, chiral, and armchair CNTs on the mechanical properties. To overcome such limitations of experimental procedures and avoid large error bars, computational simulations have been regarded as a powerful tool for accurately predicting the mechanical properties of CNTs. Using an atomistic-based model, several theoretical studies have attempted to accurately predict the mechanical properties of nanomaterials and individual CNTs [24,34,35,36,37,38,39]. These studies demonstrated that theoretical simulations can effectively investigate and predict the influence of the CNT structures on the stress–strain responses of the materials.

The structures of SWCNTs can be regarded as graphitic cylinders formed by rolling graphene sheets. A schematic of unrolled hexagonal graphene sheet is illustrated in Figure 4. The atomic structure of CNTs can be indexed by a pair of integers (*n*, *m*) corresponding to a lattice vector $\vec{C}=n{\vec{a}}_{1}+m{\vec{a}}_{2}$ on the graphite plane, where ${\vec{a}}_{1}$ and ${\vec{a}}_{2}$ denote the unit vectors of the two-dimensional hexagonal lattice. *n* and *m* are called the chiral index and are integers satisfying 0 ≤ |*m*| ≤ *n*. The integers (*n*, *m*) uniquely determine the size of SWCNTs. The symmetry groups of carbon nanotubes are represented as the zigzag tube (*m* = 0) when *θ_c_* = 0° and the armchair tube (*m* = *n*) when |*θ_c_*| = 30°.

In contrast, the chiral tube exhibits a general direction with 0 < |*θ*| < 30° or *n* > *m* > 0. As depicted in Figure 5, the tensile stress–strain responses of the three distinct CNT structures were obtained according to the continuum elasticity theory. As observed, the axial tensile stresses exhibited nonlinear dependence as a function of strain, and the elastic modulus was independent of the helicity of CNTs. According to this simulation, the stress–strain curve could be well-fitted by a binomial expression, $\sigma =A\varepsilon +B\varepsilon^{2}$ (in TPa) [37]. The linear elastic property of the CNTs was independent of the helicity of the hexagonal carbon lattice along the tubes, whereas their nonlinear elastic behavior exhibited a stronger dependence on chirality. Al-Kharusi et al. [39] conducted an atomistic finite element analysis to evaluate the energy stored in the SWCNT. In the finite element model, the force interaction between the carbon atoms in an SWCNT was modeled using load-carrying structural beams.

#### 2.1.2. Vibrations, Buckling, and Modeling

The field of miniature mechanical oscillators is rapidly evolving with emerging nanomaterials, simulation, and process technologies. Owing to the miniaturization of mechanical oscillators to the molecular scale, such as in CNT resonators, their vibrations are increasingly coupled and interact strongly, even under feeble thermal fluctuations [40,41,42,43,44]. Barnard et al. reported the direct real-time measurements of thermal vibrations of a CNT using a high-fidelity micrometer-scale silicon nitride optical cavity as a sensitive photonic microscope [40]. They discovered a realm of dynamics undetected by previous time-averaged measurements and room-temperature coherence, which were almost three orders of magnitude longer than that reported earlier. Their experimental research pioneered the study of nonlinear mechanical systems in the Brownian limit (i.e., when a system is driven solely by thermal fluctuations). Subsequently, several studies reported that CNT electromechanical resonators exhibited unprecedented sensitivity for detecting small masses and forces, which establishes the advantages of using CNTs as a nanosensor for such applications. Willick et al. reported the use of a cold heterojunction bipolar transistor amplifying the circuits near a device to measure the mechanical amplitudes on the microsecond timescales [42]. Sazanova et al. investigated the electrostatic actuation and detection of doubly clamped CNT oscillators [43]. They demonstrated that the resonance frequency could be widely tuned, and the devices could be used to measure extremely small forces. 

The natural frequency and buckling load of CNTs must be determined to realize the application of CNTs as nanosensing materials. However, most studies on the vibrational property and buckling instability of CNTs have been performed on theoretical modeling and simulation because of having difficulty in experimental research. As such, molecular and solid mechanics have been developed to fundamentally describe the properties of material with micro- and macrostructures. Although atomic simulation is adapted for nanomaterials with microstructures, they are limited to extremely small sizes as it requires enormous computational power. A nanoscale continuum model considering nonlocal parameters or an atomic-based continuum model coupled with atomistic models has been regarded as an effective method for simulating and predicting the mechanical properties of nanomaterials (e.g., CNTs and GSs). This review discusses the recent efforts on addressing the problem of modeling and simulation.

The Bernoulli–Euler beam theory described in Equation (1) has been applied for large aspect ratios and small deformations. In the case of short CNT beams and higher modes, the shear effect should be considered because the effects of through-thickness shear deformation and rotary inertia become extremely significant. Compared to the Euler–Bernoulli theory, the Timoshenko beam theory is more accurate, especially for high-frequency resonances of CNT beams. For the Timoshenko theory [45,46,47], the transverse deflection u(x, t) and the slope φ(x, t) caused by bending are governed by the following coupled differential equations:(6)$$-\kappa GA\left(\frac{\partial \varphi }{\partial x}-\frac{\partial^{2}u}{\partial x^{2}}\right)+p=\rho A\frac{\partial^{2}u}{\partial t^{2}},$$
(7)$$I\frac{\partial^{2}\varphi }{\partial x^{2}}-\kappa GA\left(\varphi -\frac{\partial u}{\partial x}\right)=\rho I\frac{\partial^{2}u}{\partial t^{2}},$$
where κ denotes the shear coefficient depending on the cross-section shape, and *G* represents the shear modulus of the CNT beam.

Unlike the aforementioned constitutive equation in classical elasticity, Eringen’s nonlocal elasticity theory [48,49] states that the stress at point *x* in a body depends on the strain at that point and all points of the body. The relationship between the stress σ and strain ε is expressed as $\sigma -{\left(e_{0}a\right)}^{2}\sigma^{\prime \prime }=E\varepsilon$, where a denotes the internal characteristic length, and indicates a constant suitable for each material [50]. The nonlocal elasticity theory can be advantageous for nanotechnology simulation applications because the nonlocal continuum mechanics model can consider the small-scale effect that emerges as significant in the case of micro- or nanostructures with discrete domains. 

The governing equation for the nonlocal Euler–Bernoulli beam model can be expressed as follows [50]:(8)$$EI\frac{\partial^{4}u}{\partial x^{4}}+\rho A\frac{\partial^{2}}{\partial t^{2}}\left(u-{\left(e_{0}a\right)}^{2}\frac{\partial^{2}u}{\partial x^{2}}\right)=p-{\left(e_{0}a\right)}^{2}\frac{\partial^{2}p}{\partial x^{2}}.$$

Considering the effect of the small length scale, Wang et al. [51] and Lu et al. [52] presented the governing equations for the nonlocal Timoshenko theory as follows: (9)$$EI\frac{\partial^{2}\varphi }{\partial x^{2}}+\kappa GA\left(\frac{\partial u}{\partial x}-\varphi \right)-\rho I\frac{\partial^{2}}{\partial t^{2}}\left(\varphi -{\left(e_{0}a\right)}^{2}\frac{\partial^{2}\varphi }{\partial x^{2}}\right)=0,$$
(10)$$\kappa GA\left(\frac{\partial \varphi }{\partial x}-\frac{\partial^{2}u}{\partial x^{2}}\right)+\rho A\frac{\partial^{2}}{\partial t^{2}}\left(u-{\left(e_{0}a\right)}^{2}\frac{\partial^{2}u}{\partial x^{2}}\right)=p-{\left(e_{0}a\right)}^{2}\frac{\partial^{2}p}{\partial x^{2}}.$$

According to these governing equations of the CNT beam and the given supported boundary conditions, the vibrational frequencies can be obtained using an analytical approach [53]. 

The vibration frequency ratio $\omega_{NT}/\omega_{NE}$ of CNTs is portrayed in Figure 6 as a function of length-to-diameter under various vibrational modes (*N*) [54], wherein subscripts *NT* and *TE* indicate the nonlocal Timoshenko (*NT*) beam model and nonlocal Euler (*TE*) beam model, respectively. As observed, the variations between the results obtained from *NT* and *NE* were negligible for long CNTs, i.e., *L*/*d* > 20, for the fundamental mode number (*N* = 1). Moreover, it is confirmed by the negligible shear effect for long nanotubes that the Timoshenko beam theory is more effective than the Bernoulli–Euler beam theory for short lengths because of the shear effect. 

For double-walled CNTs (DWCNTs) with hollow two-layer structures, the deflection of the nested tubes is coupled by the van der Waals (vdW) force between the inner and outer nanotubes [55]. The interaction pressure at any point between the two adjacent nanotubes can be regarded as a linear function of the jump in deflection (Δw) at that point, related as p=c·Δw, in which c denotes the intertube interaction coefficient per unit length between the inner and outer nanotubes. Furthermore, de Borbon et al. investigated the influence of the vdW interaction coefficient on the vibration behavior of DWCNTs [56]. The vdW interaction force between two carbon atoms of CNTs is expressed as the Lennard–Jones potential, which is widely used to describe the attractive interactions between nonpolar molecules. In certain studies, the vdW interaction coefficient c depends only on the nanotube radii and the carbon bond length and expressed as follows [57,58,59,60]:(11)$$c=\frac{200\;\left(2R\right)}{0.16\;a^{2}}\;\mathrm{erg}/{\mathrm{cm}}^{2},$$
where *R* denotes the inner nanotube radius, and a=0.142 nm indicates the length of the carbon–carbon bond. 

More recently, Saito et al. [59] provided new data according to which the vdW interaction coefficient was expressed as $c=320\;\left(2R\right)/0.16\;a^{2}\;\mathrm{erg}/{\mathrm{cm}}^{2}$.

He et al. [60] presented explicit formulas for the vdW interaction between any two layers of multiwalled CNTs. The vdW force was estimated by the first-order Taylor expansion with respect to the equilibrium position and stated as
(12)$$F\left(\bar{d}\right)=F\left({\bar{d}}_{0}\right)+\frac{dF\left({\bar{d}}_{0}\right)}{d\bar{d}}\left(\bar{d}-{\bar{d}}_{0}\right),$$
where $\bar{d}$ denotes the distance between the interacting atoms, and ${\bar{d}}_{0}$ indicates the initial distance between atoms of various tubes. The integration of Equation (12) over the entire nanotube yields an analytical representation of the initial pressure contribution caused by the vdW interaction and is expressed as
(13)$$c=\frac{\pi \varepsilon R_{1}R_{2}\sigma^{6}}{a^{4}}\left[\frac{1001\sigma^{6}}{3}H^{13}-\frac{1120}{9}H^{7}\right],$$
where
(14)$$H^{m}={\left(R_{1}+R_{2}\right)}^{-m}\int_{0}^{\pi /2}\frac{d\theta }{{\left(1-\varphi \cos^{2}\theta \right)}^{m/2}},\;\;\;\;\;\left(m=7,\;13\right)$$
and $\varphi =\frac{4R_{1}R_{2}}{{\left(R_{1}+R_{2}\right)}^{2}}\;,$ where $R_{1}$ and $R_{2}$ denote the inner and outer radii of DWCNTs, and σ and ε represent the vdW radius and the well depth of the Lennard–Jones potential, respectively. The vdW parameters in the Lennard–Jones potential are regarded as σ=0.34 nm and ε=2.967 meV [59].

In the analysis of buckling behavior, the differential equations of the nonlocal elasticity for CNTs subjected to an axial load can be expressed as follows [61,62,63,64,65,66]: For the Bernoulli–*Euler beam* theory,
(15)$$EI\frac{d^{4}u}{dx^{4}}+P\frac{d^{2}}{dx^{2}}\left(u-{\left(e_{0}a\right)}^{2}\frac{d^{2}u}{dx^{2}}\right)=p-{\left(e_{0}a\right)}^{2}\frac{d^{2}p}{dx^{2}},$$For the Timoshenko theory *beam* theory,
(16)$$EI\frac{d^{2}\varphi }{dx^{2}}+\kappa GA\left(\frac{du}{dx}-\varphi \right)=0,$$(17)$$\kappa GA\left(\frac{d\varphi }{dx}-\frac{d^{2}u}{dx^{2}}\right)+P\frac{d^{2}}{dx^{2}}\left(u-{\left(e_{0}a\right)}^{2}\frac{d^{2}u}{dx^{2}}\right)=p-{\left(e_{0}a\right)}^{2}\frac{d^{2}p}{dx^{2}},$$
where u(x) denotes the transverse deflection of the CNT beam, φ(x) represents the slope, and *P* denotes the internal axial compressive loading. 

The accurate and approximate solutions for the axial buckling load of CNTs under varying boundary conditions can be obtained from various approaches, such as the mathematical analysis of differential equations and the Bubnov–Galerkin method [67,68].

### 2.2. Mechanical Properties, Vibration, and Buckling Analysis of GSs

#### 2.2.1. The Tensile and Bending Mechanical Properties

GSs are a two-dimensional nanosubstance composed of a sheet of carbon atoms arranged in a regular hexagonal pattern. The GSs, the first truly two-dimensional materials, were discovered by Novoselov et al. in 2004 [69], for which they were awarded the Nobel Prize in Physics 2010. GSs have garnered considerable research interest owing to their excellent characteristics, such as large specific surface area, high strength and flexibility, and high thermal and elasticity conductivities [70,71,72,73].

Garcia-Sanchez et al. [74] determined the mechanical properties of multilayer GSs (MLGSs) suspended over trenches in silicon oxide for nanoelectromechanical systems, as depicted in Figure 7. The motion of the suspended MLGSs was electrostatically driven at resonance by applying radio frequency voltages. 

By modeling the suspended MLGSs with the finite element method, these edge eigenmodes were attributed to the result of nonuniform stress with remarkably large magnitudes (up to 1.5 GPa). AFM has been a major experimental approach for studying the mechanical properties of GSs. Using an AFM, Frank and Tanenbaum [75] measured the effective spring constants of stacks of MLGSs within five layers suspended over trenches in silicon dioxide. The elastic modulus of the MLGSs was experimentally investigated with force–displacement measurements. The spring constants ranging from 1 to 5 N/m were observed for suspended GSs. As such, the elastic modulus could be obtained by examining the various spring constants with the displacement of the suspended GSs. Lee et al. [76] examined the elastic properties and the intrinsic breaking strength of freestanding GSs by nanoindentation under AFM, reporting that defect-free GSs exhibited an elastic modulus of 1.0 TPa and a fracture strength of 130 GPa. These experimental results demonstrated that GSs could provide excellent mechanical properties, such as bending flexibility, tensile elastic modulus, and strength. However, the mechanical behavior could not easily be derived from the experiments due to nanomaterials with extremely small sizes, especially for single-layer graphene sheets. 

In the existing literature, certain numerical investigations have been conducted to investigate the mechanical characteristics of single-layer GSs [77,78,79,80]. Gao and Hao reported a comprehensive analysis of the mechanical behavior of monolayer graphene under tensile and compressive loading [77]. The failure mechanisms of zigzag and armchair GSs were investigated. The variations between the mechanical behavior of two typical graphene structures were derived from theoretical analysis, which cannot be obtained through experimental tests. The results demonstrated that the tensile failure and buckling failure of zigzag graphene were larger than those of armchair graphene. On the basis of the equivalence molecular potential energy, Natsuki et al. predicted the mechanical behaviors of boron nitride (BN) nanosheets, including the elastic modulus, shear modulus, and Poisson’s ratio [80]. The elastic modulus of the BN nanosheets was calculated to be 0.9–1.0 TPa, and the Poisson’s ratio of the BN nanosheets ranged from 0.09–0.15. Overall, the Young’s modulus and Poisson’s ratio were independent of the chirality of the BN nanosheets.

#### 2.2.2. Analysis Modeling, Buckling Instability, and Vibration Properties 

The buckling and vibration properties of GSs are essential for applying GSs as components in sensors or other nanodevices. Researchers have investigated the mechanical stability and vibration properties of GSs using multiple approaches. As controlled experiments on the nanoscale are extremely difficult, modeling and simulations constitute an effective approach for predicting the static and dynamic properties of nanomaterials. However, molecular dynamics (MD) simulations are time-consuming and expensive, especially for large-scale complexes. Comparing the results obtained from nonlocal continuum mechanics with those from MD simulations, nonlocal continuum mechanics can successfully describe the mechanical behaviors of nanomaterials [81,82,83]. In recent years, numerous studies exploring the mechanical behaviors of GSs, such as buckling instability [84,85,86] and vibration properties [87,88,89,90,91], have been performed according to the nonlocal theory.

As the nonlocal theory contains information on the forces between atoms, the stress depends on the strain at the individual point and all points of the component. According to the nonlocal continuum theory, an internal scale length is introduced into the constitutive equations. 

According to the plate theory modified by the nonlocal parameters of $\mu =e_{0}a$, the general governing equations for the free vibration of GSs with in-plane loads are expressed as follows [90,91,92,93,94]:(18)$$\begin{array}{l} D\nabla^{4}w-\left(1-\mu^{2}\nabla^{2}\right)\left(N_{x}\frac{\partial^{2}w}{\partial x^{2}}+2N_{xy}\frac{\partial^{2}w}{\partial x\partial y}+N_{y}\frac{\partial^{2}w}{\partial y^{2}}\right)\\ \;=-J_{0}\left(1-\mu^{2}\nabla^{2}\right)\frac{\partial^{2}w}{\partial t^{2}}+J_{2}\left(1-\mu^{2}\nabla^{2}\right)\left(\frac{\partial^{4}w}{\partial t^{2}\partial x^{2}}+\frac{\partial^{4}w}{\partial t^{2}\partial y^{2}}\right)+\left(1-\mu^{2}\nabla^{2}\right)p, \end{array}$$
where the bending stiffness $D=Eh^{3}/\left[12\left(1-\nu^{2}\right)\right]$, and *E*, *h*, and ν are the Young’s modulus, thickness, and Poisson’s ratio of GSs, respectively. $\nabla^{2}=\partial^{2}/\partial x^{2}+\partial^{2}/\partial y^{2}$ and $\left(N_{x},\;\;N_{xy},\;\;N_{y}\right)$, including the normal and shear directions, denote the in-plane force acting on the GSs. $J_{0}$ and $J_{2}$ represent the mass moments of inertia, defined as
(19)$$J_{0}=\int_{-h/2}^{h/2}\rho dz,\;J_{2}=\int_{-h/2}^{h/2}\rho z^{2}dz.$$

For the upper and lower layers (nanoplate-1 and nanoplate-2) of double-layered GSs (DLGSs), the pressure can be assumed to be linearly proportional to the deflection between the nanoplates. Thus, the governing differential equations can be described by the following two coupled equations:

Nanoplate-1:(20)$$\begin{array}{l} D\nabla^{4}w_{1}-\left(1-\mu^{2}\nabla^{2}\right)\left(N_{x}^{1}\frac{\partial^{2}w_{1}}{\partial x^{2}}+2N_{xy}^{1}\frac{\partial^{2}w_{1}}{\partial x\partial y}+N_{y}^{1}\frac{\partial^{2}w_{1}}{\partial y^{2}}\right)\\ \;=-J_{0}\left(1-\mu^{2}\nabla^{2}\right)\frac{\partial^{2}w_{1}}{\partial t^{2}}+J_{2}\left(1-\mu^{2}\nabla^{2}\right)\left(\frac{\partial^{4}w_{1}}{\partial t^{2}\partial x^{2}}+\frac{\partial^{4}w_{1}}{\partial t^{2}\partial y^{2}}\right)+\left(1-\mu^{2}\nabla^{2}\right)\left[c\left(w_{1}-w_{2}\right)\right]; \end{array}$$

Nanoplate-2:(21)$$\begin{array}{l} D\nabla^{4}w_{2}-\left(1-\mu^{2}\nabla^{2}\right)\left(N_{x}^{2}\frac{\partial^{2}w_{1}}{\partial x^{2}}+2N_{xy}^{2}\frac{\partial^{2}w_{1}}{\partial x\partial y}+N_{y}^{2}\frac{\partial^{2}w_{1}}{\partial y^{2}}\right)\\ \;=-J_{0}\left(1-\mu^{2}\nabla^{2}\right)\frac{\partial^{2}w_{2}}{\partial t^{2}}+J_{2}\left(1-\mu^{2}\nabla^{2}\right)\left(\frac{\partial^{4}w_{2}}{\partial t^{2}\partial x^{2}}+\frac{\partial^{4}w_{2}}{\partial t^{2}\partial y^{2}}\right)+\left(1-\mu^{2}\nabla^{2}\right)\left[c\left(w_{2}-w_{1}\right)\right], \end{array}$$
where *c* denotes the vdW interaction coefficient between the top and bottom layers. 

Golmakani et al. [95] presented the governing equations of buckling analysis of biaxially compressed DLGSs based on nonlocal elasticity theory considering shear stiffness, expressed as follows:

Nanoplate-1:(22)$$H_{55}\left(\frac{\partial^{2}w_{1}}{\partial x^{2}}+\frac{\partial \varphi_{1}}{\partial x}\right)+H_{44}\left(\frac{\partial^{2}w_{1}}{\partial y^{2}}+\frac{\partial \psi_{1}}{\partial y}\right)+\left(1-\mu^{2}\nabla^{2}\right)\left[N_{x}^{1}\frac{\partial^{2}w_{1}}{\partial x^{2}}+N_{y}^{1}\frac{\partial^{2}w_{1}}{\partial x^{2}}\right]=\left(1-\mu^{2}\nabla^{2}\right)\left[c\left(w_{1}-w_{2}\right)\right],$$
(23)$$D_{11}\frac{\partial^{2}\varphi_{1}}{\partial x^{2}}+\left(D_{12}+D_{66}\right)\frac{\partial \psi_{1}}{\partial x\partial y}+D_{66}\frac{\partial^{2}\varphi_{1}}{\partial y^{2}}-H_{55}\left(\frac{\partial w_{1}}{\partial x}+\varphi_{1}\right)=0,$$
(24)$$D_{66}\frac{\partial^{2}\psi_{1}}{\partial x^{2}}+\left(D_{21}+D_{66}\right)\frac{\partial \varphi_{1}}{\partial x\partial y}+D_{22}\frac{\partial^{2}\psi_{1}}{\partial y^{2}}-H_{44}\left(\frac{\partial w_{1}}{\partial x}+\psi_{2}\right)=0,$$

Nanoplate-2:(25)$$H_{55}\left(\frac{\partial^{2}w_{2}}{\partial x^{2}}+\frac{\partial \varphi_{2}}{\partial x}\right)+H_{44}\left(\frac{\partial^{2}w_{2}}{\partial y^{2}}+\frac{\partial \psi_{2}}{\partial y}\right)+\left(1-\mu^{2}\nabla^{2}\right)\left[N_{x}^{2}\frac{\partial^{2}w_{2}}{\partial x^{2}}+N_{y}^{2}\frac{\partial^{2}w_{2}}{\partial x^{2}}\right]=\left(1-\mu^{2}\nabla^{2}\right)\left[c\left(w_{2}-w_{1}\right)\right],\;$$
(26)$$D_{11}\frac{\partial^{2}\varphi_{2}}{\partial x^{2}}+\left(D_{11}+D_{66}\right)\frac{\partial \psi_{2}}{\partial x\partial y}+D_{66}\frac{\partial^{2}\varphi_{2}}{\partial y^{2}}-H_{55}\left(\frac{\partial w_{2}}{\partial x}+\varphi_{2}\right)=0,$$
(27)$$D_{66}\frac{\partial^{2}\psi_{2}}{\partial x^{2}}+\left(D_{21}+D_{66}\right)\frac{\partial \varphi_{2}}{\partial x\partial y}+D_{22}\frac{\partial^{2}\psi_{2}}{\partial y^{2}}-H_{44}\left(\frac{\partial w_{2}}{\partial x}+\psi_{2}\right)=0,$$
where $D_{ij\;}\left(i,\;j=1,\;2,\;6\right)$ denote the bending stiffness, and $H_{44},{\;H}_{55}$ denote the shear stiffness, considering the shear correction factor. 

The DLGSs are composed of two GS layers coupled with each other via vdW force. The vdW interaction coefficient *c* between the upper and lower layers can be obtained on the basis of the Lennard–Jones potential and expressed as follows [96]:(28)$$c=-{\left(\frac{4\sqrt{3}}{9a}\right)}^{2}\frac{24\varepsilon }{\sigma^{2}}\left[\frac{13\pi }{3}{\left(\frac{\sigma }{a}\right)}^{14}\frac{1}{{\left({\bar{z}}_{1}-{\bar{z}}_{2}\right)}^{12}}-\frac{7\pi }{3}{\left(\frac{\sigma }{a}\right)}^{8}\frac{1}{{\left({\bar{z}}_{1}-{\bar{z}}_{2}\right)}^{6}}\right],$$
where a denotes the characteristic internal length of the C–C bond, which is 0.142 nm. ε denotes the depth of the potential, and σ indicates a parameter determined from the equilibrium distance evaluated by the physical properties of the material. ${\bar{z}}_{j}=z_{j}/a\;\left(j=1,\;2\right)$, where $z_{j}$ denotes the coordinate of the *j*-th nanolaser in the thickness direction with the origin at the midplane of the GSs.

Kitipornchai et al. [97] derived an explicit formula for predicting the vdW interaction between any two sheets of a multilayered GS, which is expressed as
(29)$$c=-{\left(\frac{4\sqrt{3}}{9a}\right)}^{2}\frac{24\varepsilon }{\sigma^{2}}{\left(\frac{\sigma }{a}\right)}^{8}\left[\frac{3003\pi }{256}{\left(\frac{\sigma }{a}\right)}^{6}\overset{5}{\underset{k=0}{\sum }}\frac{{\left(-1\right)}^{k}}{2k+1}\left(\begin{array}{c} 5\\ k \end{array}\right)\frac{1}{{\left({\bar{z}}_{1}-{\bar{z}}_{2}\right)}^{12}}-\frac{35\pi }{8}\overset{2}{\underset{k=0}{\sum }}\frac{{\left(-1\right)}^{k}}{2k+1}\frac{1}{{\left({\bar{z}}_{1}-{\bar{z}}_{2}\right)}^{6}}\right].$$

For the DLGSs, the initial interlayer space was presumed as 0.34 nm, and the values of the interaction coefficients were calculated to be 108 GPa/nm. Thus, the vdW interaction hardly affected the mechanical properties, such as the buckling stability and the natural frequency, for lower-order modes of GSs, whereas the other higher-order parameters were significantly reliant on the vdW interaction. 

## 3. Analytical Modeling and Application of CNTs and GSs as Nanosensor Materials

### 3.1. CNTs and GSs Used for Nanomass Sensing Applications

Carbon-based nanomaterials, such as CNTs and GSs, are employed in numerous technical applications owing to their excellent features of high stiffness and strength with extremely small size. As reported, CNTs and GSs hold immense potential as resonators for nanoscale mass or force sensing [98,99,100,101,102]. In these experimental studies, controlling the mechanical motion of objects becomes a formidable challenge for fundamental science and engineering. Jensen et al. demonstrated a CNT-based nanomechanical resonator with an atomic mass resolution [98], which successfully measured a certain number of gold atoms placed in an ultrahigh vacuum. The detection technique was based on the nanotube radio-receiver design and relied on the unique field-emission properties of the CNTs. In principle, this device acted as a mass spectrometer with a mass sensitivity of 1.3 × 10^−25^ kg/Hz^1/2^.

Lassagne et al. [103] used extremely small CNTs with a diameter of 1 nm as a nanomechanical resonator for mass sensing. The resonance frequency was measured at room temperature and vacuum of ~5 × 10^−6^ mbar in an evaporator. In Figure 8, the schematics illustrate (a) the mass-sensing experimental setup, and the (b) measurement circuitry.

As depicted in Figure 8a, the chromium atoms were evaporated onto a nanotube resonator in the mass-sensing experiments. The experiment was performed in a Joule heating metal evaporator, and the mass of the atoms adsorbed on the CNT resonator was actuated and measured by electrostatic interactions. In Figure 8b, when a voltage $V_{g}^{ac}$ oscillating at a frequency was applied on the Si package of the wafer, an oscillating electrostatic force was generated on the CNT at the same frequency and expressed as
(30)$$F_{el}=C_{g}^{\prime }V_{g}^{dc}V_{g}^{ac}\cos\left[2\pi ft\right],$$
where $C_{g}^{\prime }$ denotes the derivative of the capacitance between the CNT and backage, and $V_{g}^{dc}$ indicates the DC voltage applied on the gate.

The $F_{el}$-induced motion $\delta_{z}$ of the CNTs was detected using a capacitive technique. The conductance of CNT depends on its electronic charge, controlled by the gate capacitance $C_{g}$. To track the high-frequency charge modulation, a mixing technique was employed by applying the voltage $V_{g}^{ac}\cos\left(2\pi ft\right)$ on the gate and $V_{SD}\cos\left[2\pi \left(f+\delta_{f}\right)t\right]$ on the source (Figure 8b). According to the measurement of a mixing current $I_{mix}$ on the drain at a frequency variation of $\delta_{f}$, the shifts of the resonance frequency were obtained as a function of the deposited mass. The $I_{mix}$ can be expressed as
(31)$$I_{mix}=\frac{1}{2}V_{SD}\frac{dG}{dV_{g}}\left[V_{g}^{ac}\cos\left(\alpha \right)+V_{g}^{dc}\;\frac{C_{g}^{\prime }}{C_{g}}\delta_{z}\cos\left(\theta_{m}-\alpha \right)\right],$$
where $dG/dV_{g}$ denotes the transconductance of the CNT, α indicates the electronic phase emerging from the electronic circuitry, and $\theta_{m}$ represents the phase between $F_{el}$ and the mechanical motion.

In particular, the shift in the mechanical resonance frequency of the CNT resonator across varying periods for 160 zg of Cr (~1860 Cr atoms) deposited onto the CNT (Figure 9a) and the shift in the resonance frequency (Figure 9b) were obtained as a function of the deposited mass.

The slope given the mass responsivity of the CNT resonator was ℛ=11 Hz/yg $\left(1\;\mathrm{yg}=10^{-24}\;g\right)$. As reported by Lassagne et al. [103], the exceptional mass responsivity (resonance frequency shift per unit mass change) can be attributed to the ultralight mass and high stiffness of the CNTs. Li and Zhu [104] proposed an optical detection technique with a sensitivity of a single atom using a surface plasmon and a double-clamped CNT resonator.

GSs with ultrathin two-dimensional nanomaterials are regarded as an ideal candidate for nanoelectromechanical resonators used as mass and force sensors because of their excellent mechanical properties, large surface area, and ultralight weight. In experimental research, Bunch et al. [105] demonstrated that GSs exhibit fundamental resonant frequencies in the megahertz range, which can be actuated either optically or electrically and detected optically using interferometry. Sharma et al. [106] reviewed the electronic applications of GS mechanical resonators for future-generation radiofrequency communications and ultrasensitive mass and temperature detection. They envisioned that the nonlinear characteristics of GS resonators can enable ultrasensitive mass detection using a GS mass spectrometer. 

### 3.2. Modeling and Analysis of CNT- and GS-Based Nanomass Sensor 

In the existing literature, the experimental results demonstrated that nanomaterials, such as CNTs and GSs, can be selected to construct versatile nanomechanical resonators. The detection mechanism of nanomass sensors is based on the resonant frequency shift occurring upon affixing a nanomass to the CNTs. The resonant frequency is sensitive to the resonator mass of a nanosensor system, including the mass of the resonator and measured mass that attached to the resonator. Therefore, the variations in the resonant frequency induced by the attached mass must be characterized. To date, several theoretical methods, such as the molecular structural mechanics method [107] and the continuum mechanics approaches [108,109,110,111,112], have been employed to predict the additional mass. Li and Chou [107] initially studied CNT-based nanomass resonators, which were assumed to be either cantilevered or bridged beams. The relationship between the resonant frequency of a CNT resonator and the attached mass was established by adopting the proposed approach of molecular structural mechanics [113]. In the research, the natural frequency was obtained by simulating a CNT as an equivalent space frame-like structure and solving the motion equation of the system [107]. The governing equation of the motions can be expressed as follows:(32)$$\left[M\right]\left\{\ddot{y}\right\}+\left[K\right]\left\{y\right\}=\left\{0\right\},$$
where [M] and [K] denote the global mass and stiffness matrices of the analytical system, respectively, and $\left\{\ddot{y}\right\}$ and {y} represent acceleration vector and the nodal displacement vector, respectively. 

Moreover, applying the static condensation method, the fundamental frequencies and mode shapes were obtained from the solution of the eigenproblem and expressed as
(33)$$\left({\left[K\right]}_{s}-\omega^{2}{\left[M\right]}_{s}\right)\left\{y_{p}\right\}=\left\{0\right\},$$
where ${\left[K\right]}_{s}$ and ${\left[M\right]}_{s}$ denote the condensed stiffness matrix and mass matrix, respectively, $\left\{y_{p}\right\}$ represents the displacement of the carbon atoms, and ω=2πf indicates the angular frequency.

Using the molecular structural mechanics method, the variations in resonant frequency, Δf, can be evaluated using a tiny mass with a mass matrix of [ΔM], added to the resonator to generate a matrix [M+ΔM] with the total sensor system. The simulated results indicated that the mass sensitivity of the CNT resonator attained at least 10^−21^ g, and a logarithmic, linear relationship existed between the resonant frequency and the attached mass for a mass heavier than 10^−20^ g. Moreover, the results validated the sensitivity of the resonant frequency shifts toward both the length and the diameter of the CNTs, which can be utilized to improve the range of the nanomass measurement. Applying the continuum mechanics approach, Natsuki et al. [114,115,116,117] performed a theoretical analysis of the vibration frequency in CNT- and GS-based nanomechanical mass resonators. In addition, Shen et al. [109,110] analyzed the vibration behaviors of DWCNTs according to the nonlocal Timoshenko beam theory and those of SLGS-based nanomass sensors [118] according to the nonlocal Kirchhoff plate theory. Natsuki et al. [115] reported a vibration analysis of a nanomechanical mass sensor using rectangular DLGSs as nanoresonators. Furthermore, Patel and Joshi [119] derived a second-order differential equation governing the curved DWCNTs according to Euler beam theory and Hamilton principles. Th next section reveals the general differential equations of CNTs and GSs, which are typically used by theoretical research on mass-sensing applications.

#### 3.2.1. Modeling and Differential Equation for CNT Beams

Natsuki et al. proposed an approach for building a mass sensor based on a double-end-fixed CNT with an applied tensile load and carrying an attached concentrated mass. Using the continuum model with the Euler–Bernoulli beam, the governing equation of motion of a freely vibrating CNT under the axial tensile load *N* can be expressed as follows [114]:(34)$$EI\frac{\partial^{4}w^{*}\left(x,\;t\right)}{\partial x^{4}}-N\frac{\partial^{2}w^{*}\left(x,t\right)}{\partial x^{2}}+\rho A\frac{\partial^{2}w^{*}\left(x,t\right)}{\partial t^{2}}=0,$$
where $w^{*}\left(x,\;t\right)$ denotes the flexural deflection, and N indicates the axial tensile load acting on the CNT. 

Accordingly, the relation between the boundary conditions and the nanomass, $m_{c}$, regarded as a concentrated mass, can be expressed as
(35)$$EI\frac{\partial^{3}w_{1}\left(a,t\right)}{\partial x^{3}}-EI\frac{\partial^{3}w_{2}\left(a,t\right)}{\partial x^{3}}-m_{c}\frac{\partial^{2}w_{1}\left(a,t\right)}{\partial t^{2}}=0,$$
where $w_{1}$ and $w_{2}$ denote the vibration amplitudes on the left- and right-hand sides of the attached mass, $m_{c}$, along the CNT length position of x=a, respectively. 

Furthermore, Natsuki et al. investigated the resonant frequencies and frequency shifts of the CNTs with the attached nanomass using the nonlocal Euler–Bernoulli beam model, developed to incorporate the size effect by introducing an intrinsic length scale, which provided information regarding the forces between atoms. The governing equation of motion in continuum mechanics was obtained for a CNT beam subjected to an axial tensile load *N* and expressed as follows [120]:(36)$$EI\frac{\partial^{4}w^{*}\left(x,\;t\right)}{\partial x^{4}}-\left[1-\mu^{2}\frac{\partial^{2}}{\partial x^{2}}\right]N\frac{\partial^{2}w^{*}\left(x,t\right)}{\partial x^{2}}+\left[1-\mu^{2}\frac{\partial^{2}}{\partial t^{2}}\right]\rho A\frac{\partial^{2}w^{*}\left(x,t\right)}{\partial t^{2}}=0,$$
where $\mu =e_{0}a$ denotes the nonlocal parameter relative to the carbon material, and *a* denotes the internal characteristic length of the C–C bond, which was determined to be 0.142 nm.

The boundary condition including the term of the attached nanomass, $m_{c}$, can be expressed as
(37)$$\left[1+\frac{\mu^{2}}{EI}N\right]\frac{\partial^{3}w_{1}\left(a,t\right)}{\partial x^{3}}-\left[1+\frac{\mu^{2}}{EI}N\right]\frac{\partial^{3}w_{2}\left(a,t\right)}{\partial x^{3}}-\frac{m_{c}}{EI}\frac{\partial^{2}w_{1}\left(a,t\right)}{\partial t^{2}}=0.$$

Upon substituting the solutions of the governing equations of motion into the boundary conditions, simultaneous equations containing the vibration frequency of the CNT beam and other parameters were established. The frequency shift of the CNT-based resonators with the attached mass can be obtained from the nontrivial solution of simultaneous equations. The results demonstrated that CNT under the axial tensile load yielded higher sensitivity when used as a nanomechanical mass sensor. A higher sensitivity is more beneficial for nanomass in the order of 0.001–1.0 zg. The nonlocal parameter influenced the frequency shift of the CNT nanoresonator, indicating an increase with the attached mass, especially for larger than loaded mass of 1.0 zg. 

#### 3.2.2. Modeling and Differential Equation for GS Plate

Lei et al. [117] analyzed the vibration properties of an atomic-resolution nanomechanical mass sensor modeled by a fixed supported circular monolayer GS carrying a nanoparticle at the center. The concentrated mass attached to the circular GS induced a variation in the vibrational frequency of the GS nanoplate. The governing differential equations of GS attached concentrated mass, $m_{c}$, can be expressed in polar coordinates as follows [117]:(38)$$D\nabla_{r}^{4}w\left(r,t\right)+\left[\rho h+\frac{m_{c}\delta \left(r\right)}{2\pi r}\right]\frac{\partial^{2}w\left(r,t\right)}{\partial t^{2}}=0,$$
where *r* denotes the coordinate in the radial direction, *t* indicates the time, *w*(*r*, *t*) represents the flexural deflection of the GS, *h* and *ρ* denote the thickness and mass density of GS, respectively, and (*r*) indicates the impulse function, which is denoted as
(39)$$\delta \left(r\right)=\{\begin{array}{c} \infty ,\;\;r=0\\ 0,\;\;r\neq 0 \end{array},$$
and
(40)$$\nabla_{r}^{4}=\left(\frac{\partial^{2}}{\partial r^{2}}+\frac{1}{r}\frac{\partial }{\partial r}\right)\left(\frac{\partial^{2}}{\partial r^{2}}+\frac{1}{r}\frac{\partial }{\partial r}\right),\;\;\;D=\frac{Eh^{3}}{12\left(1-\upsilon^{2}\right)},$$
where *E* denotes the elastic modulus, and υ denotes the Poisson ratio of GS.

For GSs with clamped support and a radius of *R*, the boundary conditions can be stated as
(41)$$w\left(r,t\right)=0,\;\;\;\;\;\frac{\partial w\left(r,\;t\right)}{\partial r}=0\;\;\mathrm{at}\;\;r=R.$$

The transverse vibrations of GS with a circular plate can be solved using the Laplace transform method by considering various boundary conditions. The results revealed that GSs can be employed as nanomass sensors with higher and broader ranges of sensitivities. The mass sensitivity of GSs could attain at least 10^−24^ g, with a logarithmic, linear relationship between the vibration frequency and the attached mass > 10^−24^ g.

Zhou et al. [121] reported that an attached nanoparticle could be located at an arbitrary position of the circular GS and its size effects were described according to nonlocal elasticity theory. According to the nonlocal Kirchhoff plate theory, the governing equation of circular GS carrying a nanoparticle in the polar coordinates can be expressed as
(42)$$D\nabla_{r}^{4}w\left(r,t\right)+\left[1-\mu^{2}\nabla^{2}\right]\left[\rho h+m_{c}\delta \left(r-r_{c}\right)\delta \left(\theta -\theta_{c}\right)/r\right]\frac{\partial^{2}w\left(r,t\right)}{\partial t^{2}}=0,\;\;\;\mu =e_{0}a,$$
where
(43)$$\nabla_{r}^{4}=\left(\frac{\partial^{2}}{\partial r^{2}}+\frac{1}{r}\frac{\partial }{\partial r}+\frac{1}{r^{2}}\frac{\partial^{2}}{d\theta^{2}}\right)\left(\frac{\partial^{2}}{\partial r^{2}}+\frac{1}{r}\frac{\partial }{\partial r}+\frac{1}{r^{2}}\frac{\partial^{2}}{d\theta^{2}}\right),$$
and the mass $m_{c}$ is attached to the circular GS at the position $\left(r_{c},\;\theta_{c}\right)$. The classical circular plate with a concentrated mass is recovered when the nonlocal parameter $\mu =e_{0}a$ is set to zero.

Considering the clamped supported circular GS mass sensor, the boundary conditions can be stated as
(44)$$w\left(r,\theta ,\;t\right)=0,\;\;\;\;\;\frac{\partial w\left(r,\;\theta ,t\right)}{\partial r}=0\;\;\mathrm{at}\;\;r=R,\;$$
where R denotes the radius of the circular GS.

The frequency shift Δf versus the mass of nanoparticle $m_{c}$ with R=10 nm is displayed in Figure 10 for various values of (a) the nonlocal parameter $\mu =e_{0}a$, and (b) the nondimensional location $\left(\xi =r_{c}/R,\;0\right)$ of the attached nanoparticle. The results demonstrated that the frequency shifts of the GSs increased with the attached mass. The frequency variations were evident for an attached mass heavier than 10^−21^ g, indicating that the mass sensitivity of the GS mass sensor could increase to at least 10^−21^ g. Moreover, the frequency shift of the mass sensor diminished upon considering the effect of the nonlocal parameter (Figure 10a), which was evident for larger nanoparticles. As observed from Figure 10b, the location of the nanoparticles considerably affected the frequency shift, which increased as the attached nanoparticle approached the GS center. This finding suggested that the plate center is a crucial position where the attached nanoparticle posed the strongest effect on the circular GS mass sensor.

For quadrilateral GSs loaded by nanoparticles $\left(m_{c}\right)$ in an arbitrary position $\left(x_{0},\;y_{0}\right)$, Natsuki et al. [122] proposed an analytical model to investigate the vibration behavior of DLGSs with attached nanoparticles. The governing equations describing the vibrations of the DLGS comprise two coupled differential equations and are stated as
(45)$$D\nabla^{4}w_{1}+\left[1-{\left(e_{0}a\right)}^{2}\nabla^{2}\right]\;\left[\rho h+m_{c}\delta \left(x-x_{0}\right)\delta \left(y-y_{0}\right)\right]\frac{\partial^{2}w_{1}}{\partial t^{2}}=c\left(w_{2}-w_{1}\right),$$
(46)$$D\nabla^{4}w_{2}+\rho h\left[1-{\left(e_{0}a\right)}^{2}\nabla^{2}\right]\;\frac{\partial^{2}w_{2}}{\partial t^{2}}=c\left(w_{1}-w_{2}\right),$$
where $w_{j}\left(x,\;y,\;t\right),\;j=1,\;2$ denotes the flexural deflections of the upper ( j=1) and the lower ( j=2) sheet, and *c* indicates the vdW interaction coefficient between the upper and lower layers. 

The solution of the vibration frequency for the above-stated governing equations can be obtained by introducing a substitution of
(47)$$w_{j}\left(x,y,t\right)=Y_{j}\left(x,y\right)\;e^{i\omega t},\;j=1,\;2,$$
where $Y_{j}\left(x,y\right),\;j=1,\;2$ denotes the shape function of the deflections in the upper and lower GSs, whereas *ω* denotes the resonant frequency of the DLGS sensor.

Furthermore, the governing differential equations for the vibration of GSs were derived considering the temperature dependence, which were expressed as follows [123]:(48)$$D\nabla^{4}w+\left[1-{\left(e_{0}a\right)}^{2}\nabla^{2}\right]\;\left[\rho h+m_{c}\delta \left(x-x_{0}\right)\delta \left(y-y_{0}\right)\right]\frac{\partial^{2}w}{\partial t^{2}}=N^{t}\left[1-{\left(e_{0}a\right)}^{2}\nabla^{2}\right]\;\nabla^{2}w,$$
where $N^{T}$ denotes the load induced by the temperature fluctuations, expressed as
(49)$$N^{T}=\frac{E\alpha \Delta Th}{1-\nu },$$
α denotes the coefficient of thermal expansion, and ΔT indicates the temperature variations.

According to the proposed nonlocal elasticity theory, the simulation results suggested that the vibration of DLGSs exhibited a terahertz frequency range, which could yield a higher sensitivity than SLGSs. Note that the variation errors in the frequency shifts of GSs were ~0.4% K^−1^ for varying temperatures.

### 3.3. Modeling and Analysis of Nanoforce Sensor Based on CNTs 

The extremely small dimensions and high mechanics of CNTs enable the design of CNT-based nanoforce sensors. The detection mechanism of the CNT-based nanoforce sensor is similar to the nanomass sensors, i.e., utilizing the vibration frequency shift induced by a small force on the CNTs. However, to date, only a few studies have focused on developing nanoforce sensors using CNT-based resonators. Recently, Menacer et al. proposed a new smart nanoforce sensor based on a suspended CNT gate field-effect transistor [124,125]. The study of the proposed nanoforce sensor behavior focused primarily on the relationship between the nanotube deflection and the applied force. Accordingly, they examined the ability of CNTs for detecting the applied nanoforce by ANSYS simulator and a neural network-based model of the nanoforce sensor. The structure illustrated in Figure 11 depicts a two-dimensional view of the proposed capacitive nanoforce sensor, where two materials, silicon oxide and air gap, were used as gate dielectric of the device. The function of the applied nanoforce could be described by the evolution of the drain current characteristics while examining the influence of the capacity variations of the insulating gate on the drain current. As reported, the proposed device with an extremely small diameter of the CNTs exhibited adequate sensitivity depending on the CNT dimensions.

Natsuki et al. investigated the CNT-based nanoforce sensor to analyze the buckling stability [126] and the vibration property [67,127]. To obtain high stability and resolution, the authors proposed that the structure of CNT-based resonators was composed of a DWCNT tip with a shorter outer wall [67]. Furthermore, an analytical procedure based on the Euler–Bernoulli beam model was employed to investigate the buckling instability of the DWCNT probe with varying lengths of its inner and outer tubes. The AFM probes designed by the DWCNT tip exhibited a high buckling stability, which was important for reliably detecting the loading force. The results demonstrated that the buckling stress of the DWCNT tip was ~9 GPa, which was ~3.5 times larger than that of the SWCNT [126]. The proposed analytical model of the CNT tips exhibited a DWCNT structure with varying inner and outer lengths denoted as $L_{2}$ and $L_{1}$, respectively. According to the Bernoulli–Euler beam theory, the governing differential equations of the vibration for the inner and outer nanotubes $\;\left(0\leq x\leq L_{1}\right)$ were described by the following coupled equations [67]:(50)$$EI_{1}\frac{\partial^{4}w_{1}}{\partial x^{4}}+P\frac{\partial^{2}w_{1}}{\partial x^{2}}+\rho A_{1}\frac{\partial^{2}w_{1}}{\partial t^{2}}=c\left(w_{2}-w_{1}\right),\;\;\;$$
(51)$$EI_{2}\frac{\partial^{4}w_{2}}{\partial x^{4}}+\rho A_{2}\frac{\partial^{2}w_{2}}{\partial t^{2}}=c\left(w_{1}-w_{2}\right),$$
where the subscripts 1 and 2 denote the quantities associated with the inner and outer nanotubes, respectively, $w_{j}\left(x,\;t\right),\;\;j=1,\;2$ represent the transverse deflections of the inner and outer nanotubes, respectively, P denotes the axial load acting on the inner nanotube causing the axial buckling behavior of the DWCNTs, and c denotes the vdW interaction coefficient between the inner and outer nanotubes. 

For the exposed portion of the inner CNT $\left(L_{1}\leq x\leq L_{2}\right)$, the governing differential equation of the motion can be expressed as
(52)$$EI_{3}\frac{\partial^{4}w_{3}}{\partial x^{4}}+P\frac{\partial^{2}w_{3}}{\partial x^{2}}+\rho A_{3}\frac{\partial^{2}w_{3}}{\partial t^{2}}=0,$$
where $w_{3}\left(x,\;t\right)$ represents the transverse deflection of the inner protruding nanotube.

Furthermore, Natsuki et al. [127] proposed an analytical model considering CNT-based resonators with elastic clamping. SWCNT was considered the probe of the resonator of nanoforce sensor. The elastic-supported SWCNT was partially embedded in a medium with a length of $L_{1}$ and an exposed CNT length of $L_{2}$ (Figure 2b). The interaction between the clamping medium and CNTs was described as Whitney–Riley model characterized by spring constants $\left(k_{w}\right)$ and expressed as
(53)$$k_{w}=\frac{E_{m}L_{CNT}}{\left(1+\mu_{w}\right)r_{CNT}},$$
where $E_{m}$ and $\mu_{w}$ denote the elastic modulus and Poisson’s ratio of the elastic medium, respectively; $r_{CNT}$ and $L_{CNT}$ represent the radial of CNT and the length of the clamped part, respectively.

According to the Euler–Bernoulli beam theory, the governing differential equation for the vibration of the CNT probe with elastic support for detecting an unknown external compressive force P can be expressed as follows [127]:(54)$$EI\frac{\partial^{4}w_{1}}{\partial x^{4}}+P\frac{\partial^{2}w_{1}}{\partial x^{2}}+\rho A\frac{\partial^{2}w_{1}}{\partial t^{2}}=-k_{w}w_{1},\;\;\;0\leq x\leq L_{1},$$
(55)$$EI\frac{\partial^{4}w_{2}}{\partial x^{4}}+P\frac{\partial^{2}w_{2}}{\partial x^{2}}+\rho A\frac{\partial^{2}w_{2}}{\partial t^{2}}=0,\;\;\;L_{1}\leq x\leq L,$$
where $w_{j}\left(x,\;t\right),\;j=1,\;2$ represents the flexural deflections of the embedded and the exposed segment of the CNT probe.

Upon solving the aforementioned differential equations and using the supported boundary conditions, a set of simultaneous equations was obtained. Next, according to the nontrivial solution of the obtained simultaneous equations, the frequency variations with respect to the external compressive force could be simulated and predicted using a mathematical technique. The results suggested that the CNT could be effectively used as nanoforce sensor because the vibration frequency of CNT was highly sensitive to an axial force. As observed, the stiffness of the clamping medium is crucial for the measurement accuracy of the nanoforce sensor, especially for low stiffness of the clamping medium and small clamping length [127]. As depicted in Figure 12, the influence of the length-to-diameter ratio on the fundamental natural frequency was portrayed for the DWCNT tips subjected to varying nanoforces. The detected force varied with the CNT dimensions, such as the diameter and length, thereby signifying that a high force sensitivity could be obtained by tuning the CNT lengths according to the rate of the frequency variations. 

## 4. Concluding Remarks

Carbon-based nanomaterials exhibit exceptional mechanical behaviors, such as high elastic modulus and strength, along with terahertz frequency vibration, thereby enabling a wide range of applications in the industry of nanotechnology. CNTs/GSs are the fundamental building blocks of diverse nanosensing materials. In this review, we characterized the mechanical properties of CNTs/GSs and the detection mechanism of CNT/GS-based nanosensors. This paper reviews recent studies on CNTs/GSs used as nanomechanical resonators, primarily based on continuum mechanics approaches. However, the continuum mechanics theory considers a continuous limit of lattice models for nanostructured bodies. In the analytical modeling, the nonlocal continuum mechanics theory has been developed for the nanostructured materials because of the small-scale effect, providing a reliable method to model nanoresonators. In particular, we reviewed the approaches for modeling the mechanical behavior, vibration frequency, and buckling stability of CNTs/GSs using various theoretical methods. Therefore, it is crucial to understand the unique mechanical behaviors of CNTs/GSs, as they considerably affect the performance of CNT-based resonators or nanodevices. 

In this review of nanomaterial-based sensors, the major findings indicate that CNT/GS resonators can provide high sensitivity because of their small mass and high stiffness. For measuring the frequency shifts of CNT/GS resonators, the nanobalance technique can perform mass or force detection with nanoscale precision. However, owing to a lack of an efficient and accurate characterization technique, quantitative experimental studies are not feasible for controlling the extremely small size of nanomaterials. Currently, the experimental research on CNT/GS-based sensors and related technology is in its infancy; thus, experimental efforts are required to explore the feasibility of applications of CNTs/GS nanoresonators. Their sensing performances are predicted and evaluated primarily using theoretical models and analytical approaches. With the development of nanotechnology, new methods and approaches should be well established to reduce uncertainty in measurements. Therefore, the high-performance of CNT/GS-based sensors and nanoelectronics is expected to be realized in the near future. 

## Figures and Tables

**Figure 1 nanomaterials-13-01834-f001:**
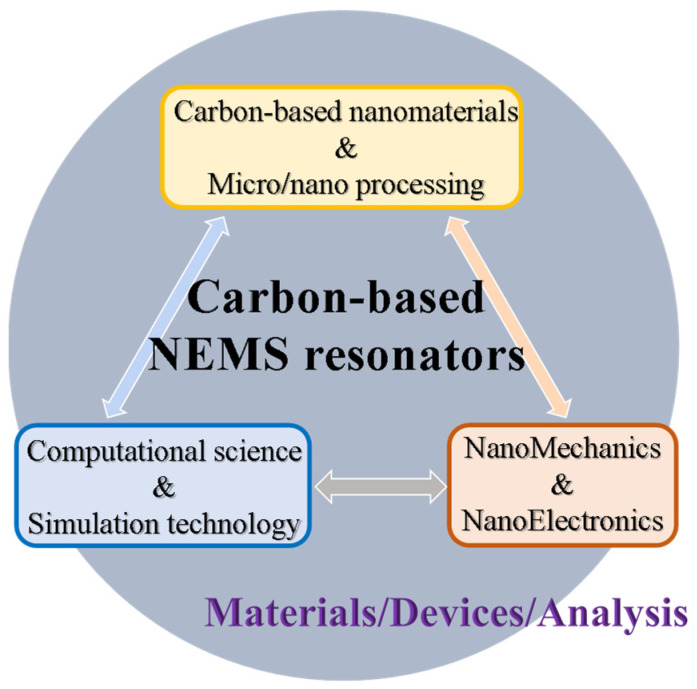
A schematic conception of carbon-based nanoelectromechanical system (NEMS) resonator integration.

**Figure 2 nanomaterials-13-01834-f002:**
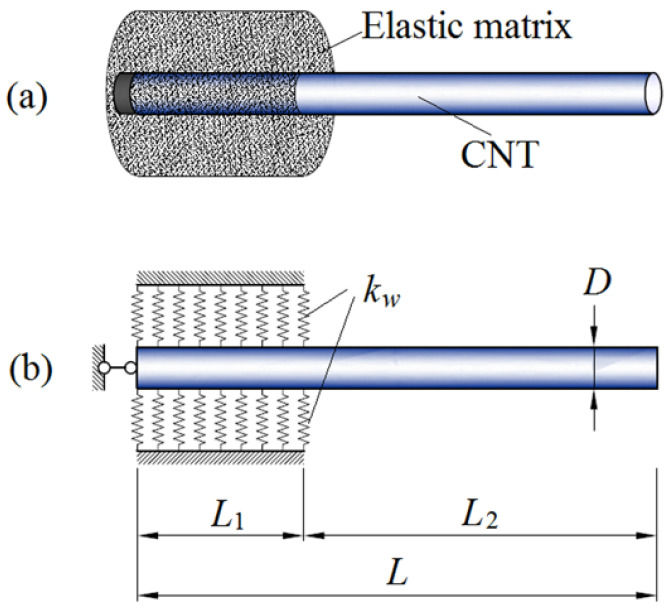
Clamped CNT beam embedded in an elastic foundation: (**a**) Schematic illustration, (**b**) Theoretical analysis model. Adapted with permission from Ref. [31]. 2018 Elsevier.

**Figure 3 nanomaterials-13-01834-f003:**
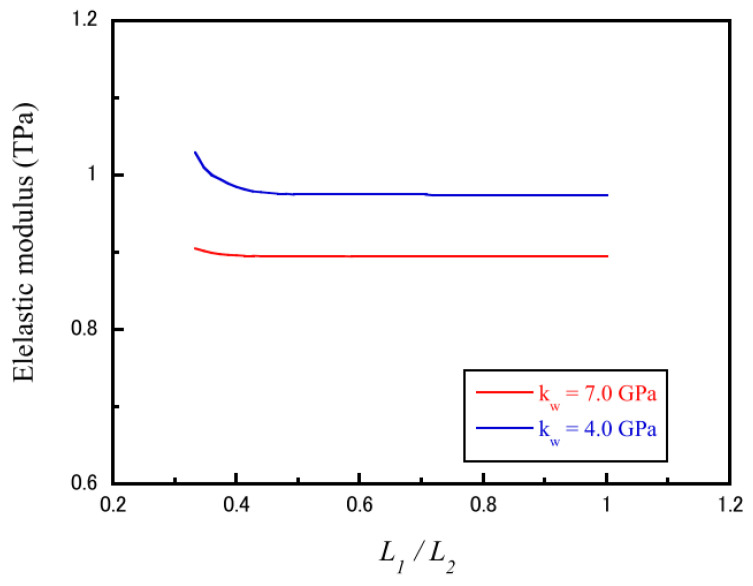
Influence of clamping stiffness on the elastic modulus of CNTs as a function of the ratio of clamped to exposed lengths. Adapted with permission from Ref. [31]. 2018 Elsevier.

**Figure 4 nanomaterials-13-01834-f004:**
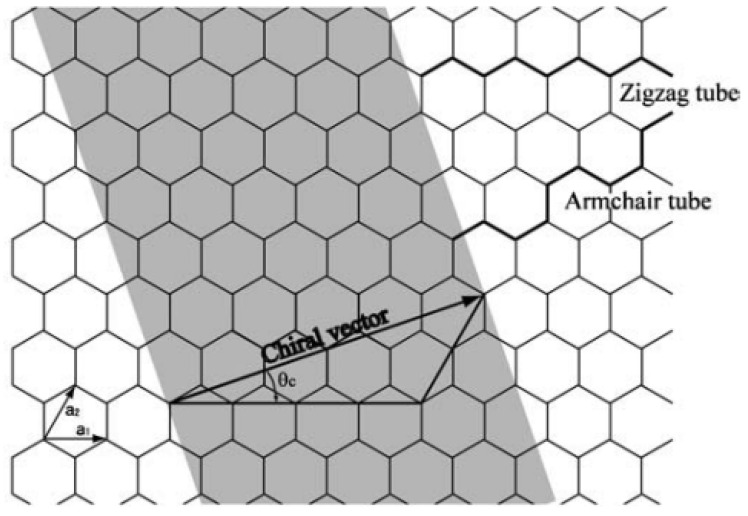
Schematic illustration of a hexagonal graphene sheet.

**Figure 5 nanomaterials-13-01834-f005:**
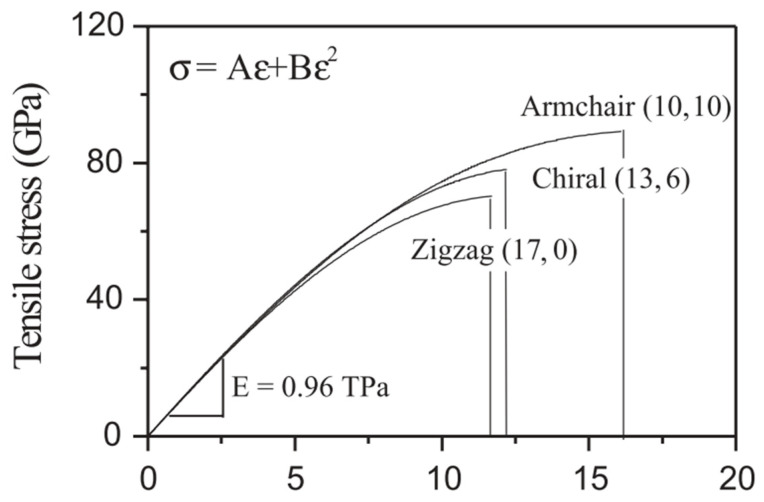
Stress–strain curves of CNTs with different helicities under tension load. Adapted with permission from Ref. [37]. 2005 Springer-Verlag.

**Figure 6 nanomaterials-13-01834-f006:**
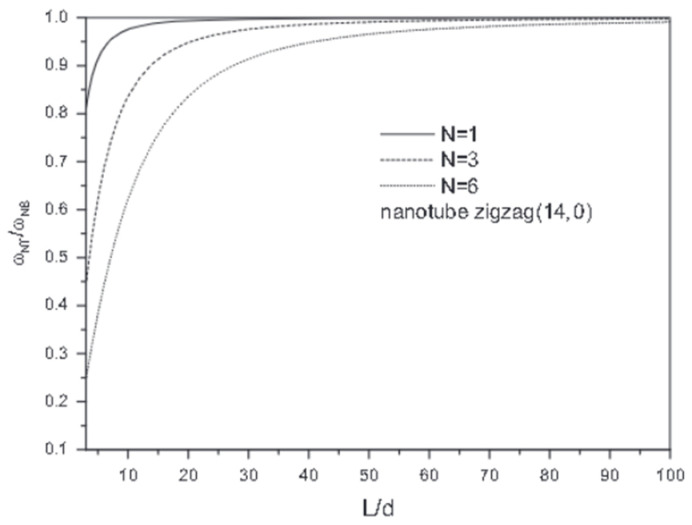
The values of frequency ratios $\left(\omega_{NT},\;\omega_{NE}\right)$ of zigzag carbon nanotube, with respect to length-to-diameter ratio for different mode numbers (*N*) using the nonlocal Timoshenko beam model (*NT*) and nonlocal Euler beam model (*NB*). Adapted with permission from Ref. [54]. 2014 Elsevier.

**Figure 7 nanomaterials-13-01834-f007:**
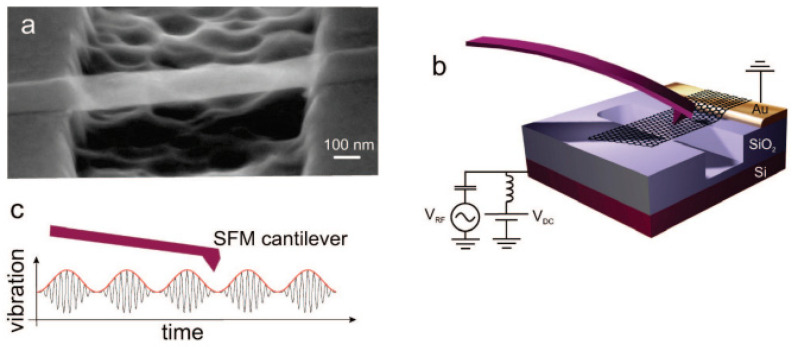
Device and experimental setup: (**a**) a scanning electron microscope image of a suspended graphene resonator; (**b**) schematic of the resonator together with the SFM cantilever; (**c**) motion of the suspended graphene sheet as a function of time. Adapted with permission from Ref. [74]. 2008 American Chemical Society.

**Figure 8 nanomaterials-13-01834-f008:**
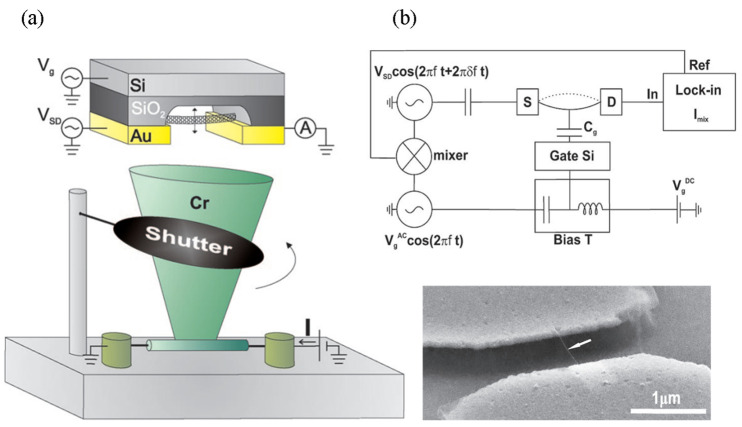
Schematic of experimental setup for mass sensing: (**a**) chromium atoms were deposited onto the nanotube resonator in a Joule evaporator, and the mass of the atoms adsorbed on the nanotube was measured; (**b**) measurement circuitry. Adapted with permission from Ref. [103]. 2008 American Chemical Society.

**Figure 9 nanomaterials-13-01834-f009:**
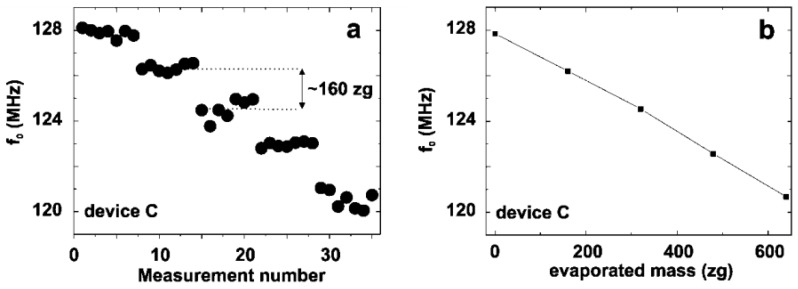
Mass-sensing experiment obtained from (**a**) resonance frequency measured sequentially, and (**b**) resonance frequency as a function of the mass evaporated on the nanotube. Adapted with permission from Ref. [103]. 2008 American Chemical Society.

**Figure 10 nanomaterials-13-01834-f010:**
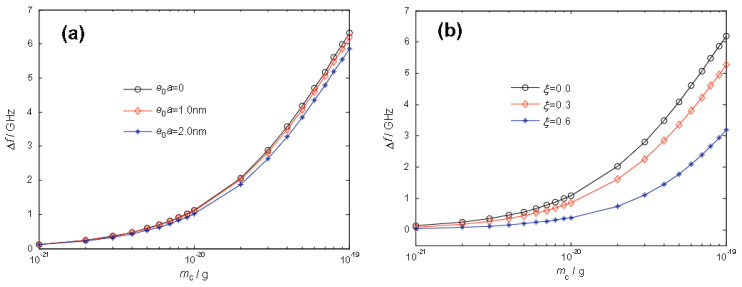
The relationship between fundament frequency shifts and the attached nanoparticles for a circular GS under simply supported boundary conditions: (**a**) the small-scale effects; (**b**) the location effects. Adapted with permission from Ref. [121]. 2014 Elsevier.

**Figure 11 nanomaterials-13-01834-f011:**
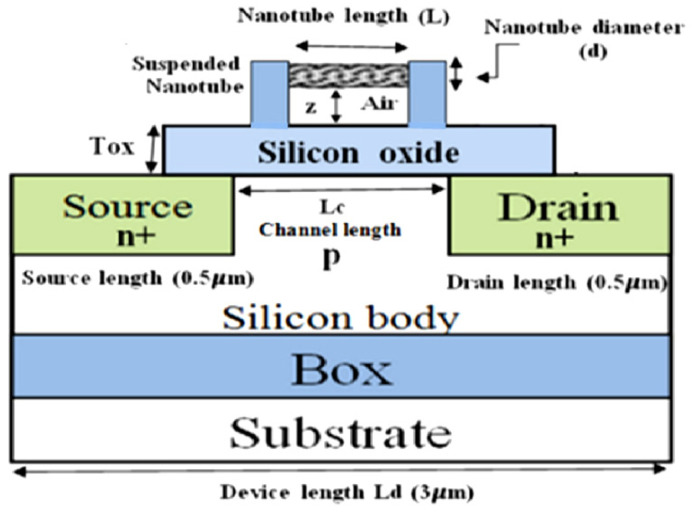
The cross-section view of the proposed capacitive nanoforce sensor. Adapted with permission from Ref. [124]. 2018 Elsevier.

**Figure 12 nanomaterials-13-01834-f012:**
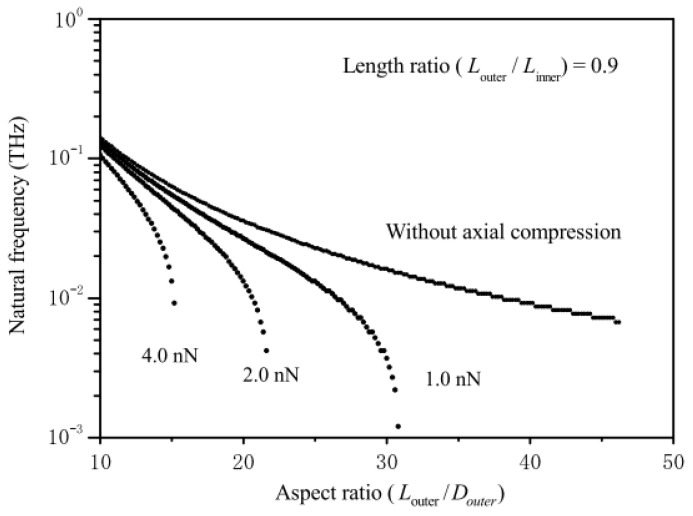
Variation of the natural frequency of cantilevered DWCNTs as the ratio of the outer tube length vs. diameter for different compressive loads. Adapted with permission from Ref. [67]. 2013 Springer-Verlag.

## Data Availability

No new data were created in this study. Data sharing is not applicable to this article.

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
