# Peer review of "Constitutive Modeling of Mechanical Behaviors of Carbon-Based CNTs and GSs, and Their Sensing Applications as Nanomechanical Resonators: A Review"

_nanomaterials, 2023, doi:10.3390/nano13121834_

Round 1

Reviewer 1 Report

This review paper describes the application of constitutive modeling and sensing of the mechanical behavior of carbon-based CNTs and GS. Carbon-based nanomaterials have been used for a variety of applications for decades due to their mechanical, physical and chemical properties. In this review paper, the authors review the development of analytical modeling of mechanical behavior with nanosensing elements based on carbon-based CNTs and GSs and their potential applications as next-generation nanosensing elements. However, I felt that the title of the paper was too comprehensive. Likewise, it was difficult to find the main keywords of this review paper in the abstract, and I was able to understand the “nano sensing element” mentioned in the abstract only after reading the main text. Therefore, the authors need to mention specifically what nanosensing is in the abstract. According to this, Figure 1 should be reconsidered.

Overall, there is a need to consolidate overlapping categories.

Modeling and analysis of GSs-based nanoforce sensors are also required.

Minor

Are all pictures with permission?

Sections 2.2.1 and 2.2.2 have similar titles. Is there any reason to separate them?

Eq.10 numbering is redundant.

Eq.29 expression is missing.

Style

Redesign the reference section style.

Use a formula font that conforms to the MDPI style.

Extensive editing of English language required

Author Response

Reviewer 1

This review paper describes the application of constitutive modeling and sensing of the mechanical behavior of carbon-based CNTs and GS. Carbon-based nanomaterials have been used for a variety of applications for decades due to their mechanical, physical and chemical properties. In this review paper, the authors review the development of analytical modeling of mechanical behavior with nanosensing elements based on carbon-based CNTs and GSs and their potential applications as next-generation nanosensing elements. However, I felt that the title of the paper was too comprehensive. Likewise, it was difficult to find the main keywords of this review paper in the abstract, and I was able to understand the “nano sensing element” mentioned in the abstract only after reading the main text. Therefore, the authors need to mention specifically what nanosensing is in the abstract. According to this, Figure 1 should be reconsidered.

Overall, there is a need to consolidate overlapping categories. Modeling and analysis of GSs-based nanoforce sensors are also required.

Answer:

Thanks for the reviewer's comments.

According to the suggestion, we more clarified the specific sensor in the title, which is “…sensing applications used as nanomechanical resonator”.

Also, we add the main keywords “nano sensing element” in the revised manuscript, and explain the “nano sensing element”.

In Lines 11-13, Nanosensors are sensing devices with sensing elements made of nanomaterials or nanostructures. CNTs- and GSs-based nanomaterials have been proved to be very sensitive nano sensing elements, detecting the tiny mass and force.

In the review, we focused on the “techniques for Carbon-based NEMS”, so, Figure 1 is revised.

Up now, Modeling and analysis of GSs-based nanoforce sensors has not been reported.

Minor

Are all pictures with permission?

Answer:

Yes, the quoted all pictures have been with permission.

Sections 2.2.1 and 2.2.2 have similar titles. Is there any reason to separate them?

Answer:

Section 2.2.1. corrected “The tensile and bending mechanical properties”.

Eq.10 numbering is redundant.

Eq.29 expression is missing.

Answer:

Sorry for our mistake, we corrected them.

Reviewer 2 Report

1. Some of the figures such as figure 11 are not up to production standard. Suggest replacing them with higher resolution images.

2. It is not clear how the test setup and samples are fabricated. Suggest adding a section discussing the sample fabrication and preparation.

3. Since this manuscript is about the use of CNT and carbon based material for sensing, suggest discussing their use in printed electronics and various sensing application. Suggest citing and discussing the article below in the introduction to enrich the discussion.

a. S. Agarwala, G. L. Goh, and W. Y. Yeong: 'Aerosol jet printed pH sensor based on carbon nanotubes for flexible electronics', Proceedings of the 3rd International Conference on Progress in Additive Manufacturing (PRO-AM), Singapore, 2018, 88-94. 

b.Chen, K., Gao, W., Emaminejad, S., Kiriya, D., Ota, H., Nyein, H. Y. Y., ... & Javey, A. (2016). Printed carbon nanotube electronics and sensor systems. Advanced Materials28(22), 4397-4414.

c. Wang, J., & Musameh, M. (2004). Carbon nanotube screen-printed electrochemical sensors. Analyst129(1), 1-2.

Nil

Author Response

Reviewer 2

  1. Some of the figures such as figure 11 are not up to production standard. Suggest replacing them with higher resolution images.

Answer:

The original author's figure does not with high resolution images.

  1. It is not clear how the test setup and samples are fabricated. Suggest adding a section discussing the sample fabrication and preparation.

Answer:

How to fabricate the test setup and samples were unknown and not reported in literatures.

  1. 3. Since this manuscript is about the use of CNT and carbon based material for sensing, suggest discussing their use in printed electronics and various sensing application. Suggest citing and discussing the article below in the introduction to enrich the discussion.
  2. S. Agarwala, G. L. Goh, and W. Y. Yeong: 'Aerosol jet printed pH sensor based on carbon nanotubes for flexible electronics', Proceedings of the 3rd International Conference on Progress in Additive Manufacturing (PRO-AM), Singapore, 2018, 88-94. 

b.Chen, K., Gao, W., Emaminejad, S., Kiriya, D., Ota, H., Nyein, H. Y. Y., ... & Javey, A. (2016). Printed carbon nanotube electronics and sensor systems. Advanced Materials, 28(22), 4397-4414.

  1. Wang, J., & Musameh, M. (2004). Carbon nanotube screen-printed electrochemical sensors. Analyst, 129(1), 1-2.

Answer:

Thanks for the reviewer's comments and suggestion.

According to the reviewer’ suggestions, we added the two suggesting citations [14,15] (Line 54, p.2) and the explanation. 

“The fabrication, and evaluation of CNT-derived screen-printed (SP) electrochemical sensors based on a CNT ink were reported. The printed CNT-based flexible electronics allowed for implementation of sensing, and a promising candidate for such printing processes which can offer stable devices with high performance.”(see Line 54-57, p.2)

Reviewer 3 Report

In this review, the authors used modeling and experiment research alongside to explain the fundamentals of nanosensor based on carbon nanotubes an graphene sheets. The literature search and presentation is comprehensive and thus I recommend it be accepted after minor revision, and my comments are listed below:

1) The author introduced the structure of GS and CNT at Page 6. In my opinion it would be discussed as early as in section 1 or the beginning of section 2.

2) In the beginning of the paper, it would be better to introduce a section to discuss the different structures of the nanosensors and their working mechanism. Also please dedicate one or several paragraphs to summarize and discuss the performance parameters of nanosensors. These would be good for readers who are new to this field.

Author Response

Reviewer 3

In this review, the authors used modeling and experiment research alongside to explain the fundamentals of nanosensor based on carbon nanotubes an graphene sheets. The literature search and presentation is comprehensive and thus I recommend it be accepted after minor revision, and my comments are listed below:

  • The author introduced the structure of GS and CNT at Page 6. In my opinion it would be discussed as early as in section 1 or the beginning of section 2.

Answer:

   We think that specific structures of GS or CNT should be introduced in the section. The CNT structures of armchair, zigzag and chiral affect their mechanical properties shown in Figure 5 in the section. In the beginning section, we did not discuss the CNT structure with spiral angle.

2) In the beginning of the paper, it would be better to introduce a section to discuss the different structures of the nanosensors and their working mechanism. Also please dedicate one or several paragraphs to summarize and discuss the performance parameters of nanosensors. These would be good for readers who are new to this field.

Answer:

Very thanks for the reviewer's comments and suggestion.

Nanosensors based on nanomaterials can be classified as silicon, carbon and so on. Different types of nanosensors such as nano-chemical sensors, electrochemical nanosensors, biosensors and nanomechanical sensor, follow different principles of detection, and various performance parameters. This is a big job for discussing sensing materials and their performance parameters. In the review, we focused mainly on the understanding of carbon-based nanomaterials using as nano sensing element and nanomechanical sensor. In the revised, therefore, we more clarified the specific sensor in the revied title, which is “…sensing applications used as nanomechanical resonator”.

Round 2

Reviewer 1 Report

All issues have been solved.

Minor editing of English language is required.